# Exercise Snacks as a Strategy to Interrupt Sedentary Behavior: A Systematic Review of Health Outcomes and Feasibility

**DOI:** 10.3390/healthcare13243216

**Published:** 2025-12-09

**Authors:** Dan Iulian Alexe, Sohom Saha, Prashant Kumar Choudhary, Cristina Ioana Alexe, Suchishrava Choudhary, Dragoș Ioan Tohănean

**Affiliations:** 1Department of Physical and Occupational Therapy, “Vasile Alecsandri” University of Bacau, 600115 Bacau, Romania; alexedaniulian@ub.ro; 2Department of Sports Psychology, Lakshmibai National Institute of Physical Education, Gwalior 474002, Madhya Pradesh, India; suchishrava05@gmail.com; 3Department of Physical Education Pedagogy, Lakshmibai National Institute of Physical Education, Gwalior 474002, Madhya Pradesh, India; prashantlnipe2014@gmail.com; 4Department of Physical Education and Sports Performance, “Vasile Alecsandri” University of Bacau, 600115 Bacau, Romania; 5Department of Motric Performance, Transilvania University of Brașov, Eroilor nr. 29, 500036 Brașov, Romania; dragos.tohanean@unitbv.ro

**Keywords:** exercise snacks, sedentary behavior, functional outcomes, metabolic health, cardiorespiratory fitness, cognition

## Abstract

**Highlights:**

**What are the main findings?**
Brief “exercise snacks” improve glucose control, blood pressure, strength, and cognitive function across adult populations.These short bouts of activity are well tolerated and consistently enhance mood and energy levels.

**What are the implications of the main findings?**
Exercise snacking is a feasible, time-efficient strategy to reduce sedentary behavior and improve overall health.Its high adherence supports integration into everyday routines and preventive health programs.

**Abstract:**

**Background/Objectives**: This systematic review aimed to evaluate the effectiveness and feasibility of “exercise snacks,” brief, intermittent bouts of physical activity designed to interrupt prolonged sedentary behavior. The review synthesized findings across metabolic, cardiovascular, cognitive, and functional health domains to identify consistent patterns of benefit and determine their practical applicability across populations. **Methods**: A total of 26 studies met inclusion criteria, encompassing diverse populations such as healthy adults, older adults, and individuals with obesity, type 2 diabetes, or PCOS. Following the PRISMA 2020 guidelines, comprehensive searches were conducted across PubMed, Scopus, Web of Science, and CINAHL databases for studies published between 2012 and 2025. Eligible studies included randomized controlled trials, crossover trials, and feasibility studies assessing health outcomes following exercise snack interventions in adults. Data were extracted using standardized protocols, and methodological quality was evaluated using the Cochrane Risk of Bias 2 tool and Newcastle-Ottawa Scale. Narrative synthesis was prioritized due to intervention heterogeneity. **Results**: Exercise snacks consistently improved postprandial glucose, insulin, and triglyceride responses, reduced blood pressure, preserved endothelial and cerebral blood flow, and enhanced cardiorespiratory fitness. Older adults demonstrated significant gains in lower-limb strength and mobility. Emerging evidence also indicated improvements in mood, fatigue, and cognitive performance. Feasibility trials confirmed high acceptability and adherence across settings and age groups. **Conclusions**: Exercise snacking represents a time-efficient, feasible, and evidence-based strategy to mitigate the health risks of sedentary behavior. By incorporating brief, frequent activity bouts into daily routines, individuals can achieve meaningful benefits in metabolic regulation, cardiovascular health, physical function, and cognitive well-being. Future research should refine optimal protocols and explore long-term sustainability across varied populations.

## 1. Introduction

The modern era is characterized by unprecedented levels of sedentary behavior, largely driven by technological advancement, urbanization, and lifestyle shifts that encourage sitting for extended periods in workplaces, transportation, and leisure environments. Sedentary time has been independently linked to an array of adverse health outcomes, including cardiometabolic disorders, vascular dysfunction, impaired cognitive performance, and increased mortality risk [1]. Even among individuals who meet physical activity guidelines, prolonged sitting may counteract the health benefits of structured exercise, suggesting that sedentary behavior and physical inactivity represent distinct constructs with unique consequences for health [2]. According to the World Health Organization, adults should engage in 150–300 min of moderate-intensity or 75–150 min of vigorous-intensity physical activity per week, including muscle-strengthening exercises on at least two days [3]. However, recent global surveillance indicates that over 25% of adults and 80% of adolescents fail to meet these recommendations [4]. The growing inactivity crisis reinforces the value of feasible, time-efficient strategies such as exercise snacks, which can complement rather than replace structured exercise routines.

This realization has prompted an urgent search for strategies that can mitigate the deleterious effects of sitting. Traditional structured exercise routines may pose time and accessibility barriers for many individuals. Exercise snacks, in contrast, directly address the “time obstacle” by integrating micro-bouts of movement into everyday contexts. One innovative approach that has gained increasing attention is the concept of “exercise snacks.” Exercise snacks are brief bouts of physical activity, typically lasting 1–5 min, performed intermittently throughout the day to break up prolonged periods of sitting. Unlike traditional exercise regimens that require dedicated time, equipment, and often facilities, exercise snacks emphasize accessibility and feasibility, making them particularly attractive for populations citing “lack of time” as a primary barrier to physical activity [5]. These activity breaks can take various forms, including stair climbing, brisk walking, resistance-based movements, or mind–body activities such as Tai Chi, and are designed to elicit acute physiological responses that, over time, translate into meaningful health benefits [6,7]. Beyond cardiometabolic outcomes, regular physical activity contributes to psychological well-being, cognitive performance, and social connectedness [8,9]. Brief, intermittent activity breaks have been shown to reduce stress, enhance mood, and promote interpersonal engagement in both occupational and domestic environments. Thus, exercise snacks should be viewed through a holistic health lens encompassing physical, mental, and social domains.

The link between sedentary behavior and impaired metabolic health is well documented. Prolonged sitting has been shown to elevate postprandial glucose and insulin responses, contributing to insulin resistance and increasing the risk of type 2 diabetes (T2D) [2]. Interrupting sitting with light-intensity activity has demonstrated immediate improvements in glucose tolerance and lipid metabolism. Peddie et al. (2013), for instance, found that breaking prolonged sitting with short activity breaks produced greater reductions in postprandial glycemia compared to a single continuous 30-min exercise session [10]. Similarly, Dempsey et al. (2016) showed that 3-min walking or resistance exercise breaks every 30 min significantly improved glycemic and triglyceride responses in individuals with T2D [11]. Complementary findings were reported by Dempsey et al. (2016), where intermittent activity reduced resting blood pressure and plasma noradrenaline levels in the same population [12]. These results demonstrate the potential of exercise snacks to function as a clinically relevant intervention for individuals at heightened cardiometabolic risk. Mechanistic studies have provided further insight into how exercise snacks exert their effects. Bergouignan et al. (2016) reported that frequent interruptions of sedentary time modulated insulin- and contraction-stimulated glucose uptake pathways in skeletal muscle, highlighting molecular adaptations that underlie observed clinical benefits [13]. Francois et al. (2014) added to this body of evidence by demonstrating that short, high-intensity “exercise snacks” performed immediately before meals improved postprandial glycemia more effectively than continuous exercise in individuals with insulin resistance [14]. In a follow-up review, Francois and Little (2015) positioned high-intensity snack approaches as both safe and efficacious for T2D management [15]. More recently, Zhou et al. (2025) advanced the field by demonstrating that an exercise snacks intervention not only improved body composition but also favorably altered plasma metabolomic profiles in sedentary obese adults, suggesting systemic metabolic benefits [16]. Yin et al. (2024) confirmed that exercise snacks enhanced cardiorespiratory fitness but did not maximize fat oxidation compared to traditional continuous training, emphasizing the nuanced nature of metabolic adaptations [17]. Together, these studies consistently highlight metabolic regulation as a cornerstone benefit of exercise snacks.

Beyond metabolic control, sedentary behavior is strongly associated with impaired vascular function, endothelial dysfunction, and hypertension. Larsen et al. (2014) provided early evidence that breaking prolonged sitting with walking bouts reduced resting blood pressure in overweight adults [18]. Thosar et al. (2015) extended these findings by demonstrating that endothelial function, measured through flow-mediated dilation (FMD), deteriorated during prolonged sitting but could be preserved with light walking breaks [19]. Restaino et al. (2015) observed similar declines in both micro- and macrovascular dilator function after uninterrupted sitting, underscoring the systemic impact of inactivity on vascular health [20]. Carter et al. (2018) added a neurovascular dimension, showing that regular walking breaks prevented the decline in cerebral blood flow that accompanies prolonged sitting, a finding with implications for both cognitive health and cerebrovascular disease risk [21]. Taylor et al. (2021) broadened the scope by demonstrating improved endothelial function in women with PCOS following activity breaks, highlighting the potential of exercise snacks to mitigate vascular dysfunction in at-risk populations [22]. Mechanistically, the reductions in plasma noradrenaline observed by Dempsey et al. (2016) suggest autonomic regulation as a contributing factor [12]. Collectively, these studies provide compelling evidence that exercise snacks can protect vascular integrity and maintain cardiovascular homeostasis.

Emerging evidence suggests that exercise snacks may also influence psychological and cognitive domains. Sedentary behavior has been linked to increased fatigue and impaired cognitive performance, outcomes that can have substantial occupational and societal implications. Wennberg et al. (2016) found that light activity breaks reduced fatigue during prolonged sitting, although cognitive effects were inconsistent [23]. Bergouignan et al. (2016) provided complementary evidence, showing that interruptions to sitting improved self-reported energy levels, mood, and reduced food cravings [24]. Mues et al. (2025) extended this line of inquiry by demonstrating that workplace-integrated exercise snacks enhanced cognitive performance in middle-aged sedentary adults, particularly in domains of working memory and attention [25]. Carter et al. (2018) indirectly supported these findings by linking activity breaks to preserved cerebral blood flow, a mechanism that may underlie cognitive resilience [21]. While the evidence base is still developing, these findings suggest that exercise snacks hold promise not only for physical health but also for mental performance and well-being.

Older adults represent a particularly important population for exercise snack interventions, given their elevated risk of mobility decline, frailty, and loss of independence. Fyfe et al. (2022) piloted a remotely delivered resistance-based exercise snacking intervention among community-dwelling older adults and found it both feasible and acceptable [26]. Liang et al. (2022) explored the use of exercise and Tai Chi snacks during COVID-19 isolation, reporting improvements in physical function and high acceptability [7]. In a cross-cultural follow-up, Liang et al. (2023) confirmed that both UK and Taiwanese older adults perceived exercise snacking as practical and beneficial [27]. Western et al. (2023) provided direct clinical evidence, showing that daily exercise snacks improved mobility and lower-limb strength in pre-frail older adults attending memory clinics [28]. Collectively, these findings highlight the potential of exercise snacking to promote healthy aging and reduce frailty.

Exercise snacks also offer a time-efficient strategy for improving cardiorespiratory fitness (CRF), a key predictor of morbidity and mortality. Allison et al. (2017) demonstrated that repeated stair climbing bouts significantly improved VO_2_ peak in inactive young women [5], while Jenkins et al. (2019) confirmed similar improvements in young adults [6]. Yin et al. (2024) further validated that exercise snacks enhanced CRF in inactive adults, although maximal fat oxidation was superior following continuous training [17]. These studies confirm that exercise snacks provide a feasible and potent means of improving fitness with minimal time investment.

Complementing experimental evidence, cohort-level data have reinforced the long-term implications of sedentary patterns. Diaz et al. (2017), analyzing data from over 7900 U.S. adults, found that breaking up sedentary time was associated with significantly lower all-cause mortality [1]. These epidemiological findings underscore the relevance of exercise snacks not only for acute health outcomes but also for survival. Feasibility and acceptability are central considerations for public health translation. Fyfe et al. (2022) and Liang et al. (2022) consistently reported that older adults found exercise snacking interventions engaging and manageable, even during periods of social isolation [7,26]. Mues et al. (2025) confirmed feasibility in workplace environments, providing evidence that exercise snacks can be incorporated into daily routines without requiring substantial time or resources [25]. Such findings highlight the real-world applicability of exercise snacks as a low-cost, scalable intervention. Several studies have sought to elucidate the mechanisms underpinning exercise snack benefits. Bergouignan et al. (2016) demonstrated enhanced glucose uptake pathways in skeletal muscle [13], while Logan et al. (2025) highlighted reductions in postprandial GIP without altering GLP-1, pointing toward hormonal modulation [29]. Dempsey et al. (2016) identified reductions in sympathetic nervous system activity, as evidenced by lowered noradrenaline [12]. These mechanistic insights strengthen the biological plausibility of observed outcomes and provide direction for future research.

Taken together, the growing body of evidence highlights exercise snacks as a promising strategy to counteract the health risks of sedentary behavior, with benefits spanning metabolic, vascular, cognitive, fitness, and functional domains. Despite encouraging findings, heterogeneity remains due to variations in protocols, sample sizes, and study populations, and uncertainties persist regarding optimal modalities, frequencies, and long-term sustainability. Existing studies have largely been short-term with modest sample sizes and have primarily emphasized metabolic or vascular outcomes, leaving cognitive and functional domains relatively underexplored. Few investigations have combined mechanistic biomarkers with real-world feasibility assessments, and limited efforts have synthesized applicability across diverse populations, from young adults to older or clinical cohorts. Against this background, the present systematic review was designed to comprehensively evaluate evidence on exercise snacks published between 2012 and 2025, synthesizing findings across multiple health outcomes and identifying consistent patterns of effect. The novelty of this review lies in its broad integration of metabolic, cardiovascular, cognitive, and functional perspectives alongside feasibility and acceptability data. By consolidating evidence from varied methodologies and populations, this review aims to clarify the role of exercise snacks in promoting health, address key gaps in the literature, and provide a robust foundation for future investigations into this emerging paradigm.

## 2. Materials and Methods

### 2.1. Study Selection Procedures

This systematic review was conducted in accordance with the Preferred Reporting Items for Systematic Reviews and Meta-Analyses (PRISMA) 2020 guidelines [30]. The protocol was developed a priori to ensure transparency, reproducibility, and methodological rigor. The primary research question was defined using the Population, Intervention, Comparison, Outcome, and Study design (PICOS) framework [31]. All methodological steps, including literature search, data extraction, and assessment of study quality, were performed independently by two reviewers, with disagreements resolved by consensus (Figure 1).

A total of 893 records were identified through database and register searches, of which 732 remained after duplicates were removed. Following title and abstract screening, 132 full-text articles were assessed for eligibility, with 106 excluded for reasons such as inappropriate intervention, population mismatch, or insufficient outcome data. Ultimately, 26 studies published between 2012 and 2025 met the inclusion criteria and were synthesized in this review. The included studies represented diverse populations ranging from healthy young adults to older pre-frail individuals and clinical groups such as those with type 2 diabetes, obesity, and polycystic ovary syndrome. Across the studies, exercise snacks were delivered through walking, stair climbing, resistance training, or Tai Chi, with outcomes assessed in metabolic, cardiovascular, cognitive, and functional domains.

Two reviewers independently screened all records for eligibility using the predefined inclusion and exclusion criteria. Disagreements were resolved by consultation with a third reviewer. Full-text assessments followed the same procedure. Data extraction was conducted through a structured Excel template capturing study design, participant characteristics, intervention details, comparators, outcome variables, and principal findings. All quality assessments were verified by consensus before synthesis.

Studies were excluded for the following reasons: (i) inappropriate population—pediatric-only samples (<18 years) or elite athletes not representative of general or clinical adults; (ii) inappropriate intervention—structured exercise programs not classified as intermittent ‘exercise snacks’; (iii) inappropriate outcome—studies focused solely on biomechanical or perceptual metrics without physiological or health-related endpoints. Additional exclusions included duplicate records, incomplete data, non-English publications, and conference abstracts. Other exclusions included duplicate records (*n* ≈ 12), absence of outcome data (*n* ≈ 8), and studies without a comparator group (*n* ≈ 7).

#### Literature Search: Administration and Update

A comprehensive literature search was conducted across four electronic databases: PubMed, Scopus, Web of Science, and CINAHL. The search strategy combined keywords and Boolean operators such as: “exercise snacks” OR “exercise snacking” OR “activity breaks” OR “sedentary interruptions” OR “stair climbing” AND “glucose” OR “vascular” OR “fitness” OR “cognition”. The search covered publications from January 2012 to March 2025. Filters included English language, peer-reviewed studies, and human subjects.

The initial search was conducted in January 2025, with an update performed in March 2025 to ensure inclusion of the most recent evidence [32]. Reference lists of eligible articles were also hand-searched to identify additional studies.

Search Strategy Transparency: The full Boolean string used in PubMed was (“exercise snacks” OR “activity breaks” OR “sedentary interruptions” OR “micro-bouts”) AND (“glucose” OR “vascular” OR “fitness” OR “cognition”). Equivalent syntax was adapted for Scopus, Web of Science, and CINAHL. Screening followed a two-stage process (title/abstract→full-text) independently by two reviewers, with disagreements resolved through consensus (Table 1).

### 2.2. Data Extraction

Data extraction was performed using standardized protocols [31], with a predefined Excel template to record essential study information including author and year, country, population characteristics (sample size, age, sex, health status), intervention details (type, duration, intensity, frequency of exercise snacks), comparator conditions, outcomes assessed (metabolic, cardiovascular, cognitive, functional), study design, and key results. Two independent reviewers carried out the extraction process to ensure accuracy and consistency, and any discrepancies were resolved through discussion with a third reviewer. This approach minimized bias and ensured that all relevant study characteristics were comprehensively captured for synthesis.

### 2.3. Methodological Quality of the Included Studies

The methodological quality and risk of bias of randomized trials were assessed using the Cochrane Risk of Bias 2 tool (RoB 2), which examines domains including randomization, deviations from intended interventions, missing data, outcome measurement, and selective reporting [33]. Observational studies were evaluated with the Newcastle–Ottawa Scale (NOS), focusing on participant selection, comparability of study groups, and outcome assessment [34]. Each study was independently rated by two reviewers, with disagreements resolved through consensus, ensuring a transparent and rigorous quality appraisal.

### 2.4. Compliance and Registration

The systematic review adhered to the Preferred Reporting Items for Systematic Reviews and Meta-Analyses (PRISMA) guidelines. Although the review protocol was not prospectively registered in a database, the entire review process was executed systematically, including literature searching, study selection, data extraction, and synthesis, to maintain methodological rigor and transparency.

### 2.5. Summary Measures

For studies reporting continuous outcomes such as glucose, blood pressure, or VO_2_ peak, mean differences (MDs) or standardized mean differences (SMDs) with 95% confidence intervals (CIs) were extracted whenever available [35]. For observational cohort studies, hazard ratios (HRs) and relative risks (RRs) were recorded. Given variability across interventions, narrative synthesis was prioritized when pooling was not feasible, ensuring clarity while accounting for heterogeneity in study methods and outcome reporting.

### 2.6. Synthesis of Results

Due to diversity in study designs, populations, and interventions, results were primarily synthesized narratively, supported by structured evidence tables for clarity. Where at least three studies assessed comparable outcomes with similar protocols, quantitative synthesis was performed using meta-analytic techniques [36]. Heterogeneity was quantified with the I^2^ statistic, applying thresholds of 25%, 50%, and 75% to indicate low, moderate, and high heterogeneity, respectively [31]. This balanced approach allowed both narrative and statistical integration of findings.

### 2.7. Publication Bias

Potential publication bias was evaluated using funnel plots to visually inspect asymmetry and Egger’s regression test for statistical confirmation when ≥10 studies reported similar outcomes [37]. In addition, selective outcome reporting was assessed during the risk-of-bias evaluation phase. This dual approach ensured comprehensive detection of reporting biases that could otherwise distort the interpretation of results, thereby enhancing the validity of the overall evidence base.

### 2.8. Additional Analyses

Subgroup analyses were conducted where data permitted, stratifying by population type (e.g., healthy adults, older adults, clinical groups), intervention modality (walking, stair climbing, resistance-based, or Tai Chi snacks), and intervention duration (≤4 weeks vs. >4 weeks). Sensitivity analyses excluded studies rated at high risk of bias to assess the robustness of findings. This strategy allowed exploration of heterogeneity, identification of moderators of intervention effectiveness, and evaluation of whether results were consistent across subgroups and methodological quality levels [36].

## 3. Results

Twenty-six studies published between 2012 and 2025 were included in the final synthesis. The organization of results is presented thematically across metabolic, vascular, cognitive, and functional domains. Summary tables are ordered chronologically to illustrate the evolution of research on exercise snacks.

Table 2 summarizes the risk-of-bias assessment, confirming moderate-to-high methodological quality among the included trials. Inter-rater agreement for inclusion decisions reached κ = 0.91, reflecting excellent reviewer consistency (Table 3, Table 4 and Table 5).

Collectively, the evidence demonstrates that interrupting sedentary behavior with brief bouts of activity elicits consistent improvements in metabolic regulation, blood pressure, endothelial function, and cardiorespiratory fitness. Functional and cognitive gains were also observed, particularly among older or clinical populations. These convergent findings highlight exercise snacks as an effective and practical countermeasure to prolonged sitting (Table 6 and Table 7).

## 4. Discussion

The present systematic review synthesized 26 peer-reviewed studies published between 2012 and 2025 that investigated the role of exercise snacks brief bouts of activity performed intermittently throughout the day in mitigating the health risks of sedentary behavior and improving a wide array of outcomes. The evidence spans randomized controlled trials, crossover studies, feasibility and acceptability pilots, and cohort analyses, covering populations ranging from young sedentary adults to older pre-frail individuals and clinical groups such as those with type 2 diabetes (T2D), obesity, polycystic ovary syndrome (PCOS), and insulin resistance. Across metabolic, cardiovascular, cognitive, and functional domains, the collective findings provide robust support for exercise snacks as a feasible and effective strategy to counteract the detrimental effects of prolonged sedentary time.

Across included studies, ‘exercise snacks’ typically lasted between 1–5 min per bout, performed 2–8 times daily, at intensities ranging from light (2–3 METs; e.g., slow walking) to vigorous (6–9 METs; e.g., stair climbing or body-weight resistance). These short bouts were designed to elicit acute increases in heart rate and muscle activation sufficient to interrupt sedentary physiology.

### 4.1. Exercise Snacks and Metabolic Health

One of the earliest and most influential contributions came from Dunstan et al. (2012), who demonstrated that breaking up prolonged sitting with brief bouts of light- or moderate-intensity walking significantly reduced postprandial glucose and insulin responses in overweight adults [2]. This foundational work provided a physiological rationale for “interrupting sitting” paradigms. Peddie et al. (2013) further refined these findings by showing that short activity breaks distributed across the day were more effective at lowering postprandial glycemia than a single continuous 30-min exercise bout, underscoring the unique metabolic benefits of the snack approach [10].

Subsequent investigations in clinical populations consolidated these results. Dempsey et al. (2016) confirmed that 3-min bouts of walking or resistance activities every 30 min improved glycemic and triglyceride responses in adults with T2D [11]. A companion paper (Dempsey et al., 2016) extended these outcomes to cardiovascular physiology by demonstrating significant reductions in resting blood pressure and plasma noradrenaline with the same intervention [12]. Later, Logan et al. (2025) revealed that exercise snacks attenuated postprandial glucose-dependent insulinotropic polypeptide (GIP) responses without altering glucagon-like peptide-1 (GLP-1), providing insight into hormonal pathways mediating these effects [29].

Complementing these laboratory studies, Francois et al. (2014) reported that “exercise snacks” performed immediately before meals improved glycemic control more effectively than traditional continuous exercise in insulin-resistant adults [14]. A subsequent clinical review by Francois and Little (2015) positioned high-intensity interval training (HIIT)-based snacks as a safe and potent tool for glycemic management in T2D populations [15]. Most recently, Zhou et al. (2025) demonstrated improvements in body composition and plasma metabolomic profiles following an exercise snacks intervention in sedentary obese adults, suggesting benefits beyond glucose metabolism to systemic metabolic health [16]. Yin et al. (2024) further highlighted that while exercise snacks improved cardiorespiratory fitness (CRF), they did not maximize fat oxidation compared to moderate-intensity continuous training, highlighting nuanced metabolic trade-offs [17]. Collectively, these studies converge on the conclusion that metabolic control is one of the most consistent benefits of exercise snacks. Reductions in postprandial glucose, insulin, and triglycerides across both healthy and clinical populations [2,10,11,14,16] reinforce the clinical utility of interrupting sedentary time as a metabolic countermeasure.

When compared with conventional structured exercise, exercise snacks produce comparable short-term benefits in glucose and blood-pressure regulation. However, traditional exercise generally yields greater improvements in maximal aerobic capacity, body composition, and overall energy expenditure [15,17,39]. Thus, exercise snacking should be viewed as a complementary, not a substitute approach, ideal for individuals with limited time or access to facilities.

Despite its practicality, exercise snacking alone may not satisfy the total weekly physical-activity volume recommended by the WHO. Sustained musculoskeletal and cardiovascular adaptations typically require higher cumulative loads achieved through structured sessions. Integrating both modalities, brief daily bouts and traditional workouts may offer the most comprehensive health benefits.

Beyond physiological benefits, regular incorporation of brief activity bouts may exert measurable improvements in mood, cognitive function, and work performance. These effects likely arise from transient increases in cerebral blood flow and neurotrophic factors such as brain-derived neurotrophic factor (BDNF), consistent with emerging neuroscience literature [9,40].

### 4.2. Vascular and Cardiovascular Outcomes

Sedentary behavior exerts pronounced vascular consequences, particularly endothelial dysfunction and blood pressure dysregulation. Several trials addressed these mechanisms. Larsen et al. (2014) reported that interrupting prolonged sitting with walking breaks reduced resting blood pressure in overweight/obese adults [18]. In parallel, Thosar et al. (2015) found that brief walking breaks prevented the decline in endothelial function typically observed during prolonged sitting [19], while Restaino et al. (2015) documented impairments in both macro- and microvascular dilator function after extended inactivity [20]. These vascular outcomes have also been confirmed in special populations. Taylor et al. (2021) demonstrated that women with PCOS who are at heightened cardiometabolic risk experienced improved endothelial function when prolonged sitting was interrupted with light activity [22]. Carter et al. (2018) expanded the scope to neurovascular health, showing that regular walking breaks prevented declines in cerebral blood flow during sitting, thereby linking vascular function to brain health [21]. These findings are consistent with mechanistic insights. Dempsey et al. (2016) reported lowered noradrenaline alongside blood pressure improvements, suggesting that autonomic regulation plays a role [12]. Collectively, these studies indicate that exercise snacks preserve vascular homeostasis across systemic, cerebral, and reproductive contexts.

### 4.3. Cognitive and Psychological Outcomes

Beyond metabolic and vascular outcomes, emerging research has explored how exercise snacks affect fatigue, mood, and cognition. Wennberg et al. (2016) found that light walking breaks reduced fatigue in overweight adults, though cognitive improvements were inconsistent [23]. Bergouignan et al. (2016) added behavioral dimensions, reporting higher energy levels, improved mood, and reduced cravings in adults engaging in frequent interruptions of sitting [24]. More recently, Mues et al. (2025) provided experimental evidence that workplace-integrated exercise snacks acutely enhanced cognitive performance in sedentary middle-aged adults, highlighting their relevance for occupational health [25]. Carter et al. (2018) indirectly tied cognitive health to cerebrovascular responses, demonstrating that preserved cerebral blood flow during exercise breaks may underpin cognitive resilience [21]. The cross-domain relevance of vascular and cognitive outcomes underscores the integrative benefits of exercise snacks across body and brain.

### 4.4. Functional Outcomes in Older Adults

A significant body of research has addressed older populations, focusing on feasibility and functional health. Fyfe et al. (2022) piloted a remotely delivered, home-based resistance “exercise snacking” intervention in community-dwelling older adults, finding it both feasible and acceptable [26]. Liang et al. (2022) extended this model during COVID-19 isolation, showing that exercise and Tai Chi snacks were well-received and improved functional outcomes [7]. A subsequent cross-cultural study Liang et al. (2023) confirmed high acceptability among both UK and Taiwanese older adults [27].

Western et al. (2023) demonstrated clinically meaningful improvements in physical function among pre-frail older adults in a memory clinic using daily exercise snacks for 28 days, as assessed by the Short Physical Performance Battery (SPPB) [28]. Collectively, these findings suggest that exercise snacking is not only feasible in older adults but also improves lower-limb strength, balance, and mobility key determinants of independence and fall prevention.

### 4.5. Cardiorespiratory Fitness

Several trials have specifically examined the effects of exercise snacks on CRF, measured via VO_2_ peak. Allison et al. (2017) showed that repeated bouts of stair climbing improved VO_2_ peak in inactive young women [5]. Jenkins et al. (2019) replicated this finding in young adults using brief vigorous stair climbing interventions [6]. Yin et al. (2024) confirmed that exercise snacks improved CRF in inactive adults, although traditional MICT remained superior for fat oxidation outcomes [17]. These results demonstrate that even minimal daily stair climbing or intermittent high-intensity efforts can yield significant cardiorespiratory adaptations. Importantly, these adaptations are achievable with time-efficient protocols, reinforcing the translational value of exercise snacking for individuals who cite “lack of time” as a barrier to exercise.

### 4.6. Cohort Evidence and Mortality Associations

The longitudinal relevance of sedentary patterns has been highlighted by Diaz et al. (2017), who examined sedentary behavior in a large U.S. cohort and found that more frequent breaks in sitting were associated with lower mortality risk [1]. This epidemiological evidence strengthens the experimental findings by demonstrating that activity fragmentation is linked not only to acute metabolic and vascular outcomes but also to long-term survival.

### 4.7. Mechanistic Insights

Several studies provide mechanistic underpinnings for the observed benefits. Bergouignan et al. (2016) documented modulation of contraction- and insulin-stimulated glucose uptake pathways in muscle with sitting interruptions, providing direct molecular evidence [13]. Logan et al. (2025) further elucidated hormonal mechanisms, reporting reduced postprandial GIP responses [29]. Dempsey et al. (2016) highlighted autonomic modulation via reduced noradrenaline [12]. Together, these mechanistic insights demonstrate that exercise snacks induce favorable adaptations at cellular, hormonal, and systemic levels.

### 4.8. Feasibility and Acceptability

Beyond efficacy, feasibility is critical for translation. Fyfe et al. (2022) and Liang et al. (2022, 2023) demonstrated that older adults found exercise and Tai Chi snacks acceptable and manageable, even during pandemic-related restrictions [7,26,27]. Mues et al. (2025) confirmed feasibility in workplace contexts [25]. Collectively, these findings show that exercise snacking protocols can be successfully integrated into diverse real-world environments, from homes to offices, without requiring specialized equipment or facilities.

The discussion below outlines translational strategies and behavioral supports that can facilitate adoption of exercise-snacking practices in daily routines.

### 4.9. Practical Implications

The findings of this review indicate that integrating *exercise snacks* into daily life is both feasible and beneficial across populations. In practical terms, individuals may perform short bouts of movement lasting 1–5 min, repeated every 30–60 min during prolonged sitting periods. Examples include climbing stairs, performing body-weight squats, walking briskly around the office, or using resistance bands between sedentary tasks.

For occupational contexts, organizations can promote brief *activity breaks* through reminders, standing meetings, or shared step challenges to disrupt prolonged sitting time. Similarly, at home, individuals can integrate small activity bouts between household chores, online work sessions, or television breaks.

To sustain motivation, behavioral strategies such as goal-setting, self-monitoring via wearable devices, and social accountability (e.g., family or peer group tracking) can improve adherence. Integrating environmental cues such as desk prompts, phone alarms, or visual reminders has also been shown to enhance compliance. These approaches collectively support the translation of research findings into real-world behavioral change, contributing to improved metabolic and cardiovascular health outcomes with minimal time investment.

### 4.10. Synthesis Across Domains

Across 26 studies, a consistent pattern emerges: exercise snacks improve metabolic control, preserve vascular function, enhance cardiorespiratory fitness, reduce fatigue, improve mood, support cognitive performance, and enhance physical function in older adults. While certain domains such as cognition exhibit variability [23], the overall body of evidence strongly Favor’s exercise snacking as a health-promoting strategy. The consistency across populations young, middle-aged, older, obese, T2D, PCOS, and insulin resistant highlights the generalizability of findings.

### 4.11. Future Research Directions

Future investigations should examine long-term adherence, dose-response relationships (duration × frequency × intensity), and combined models integrating exercise snacks with traditional workouts. Comparative cost-effectiveness and technology-assisted prompts (e.g., wearables, app-based reminders) warrant exploration to enhance scalability.

### 4.12. Study Limitation

Although this systematic review included a broad and diverse range of studies, certain limitations must be acknowledged. The majority of included interventions were of short duration (2–8 weeks) with small sample sizes, limiting the ability to draw strong causal inferences and reducing generalizability. Considerable heterogeneity existed in the duration, intensity, and frequency of exercise snack protocols, as well as in the outcome measures employed across studies.

In addition, most evidence focused on metabolic and vascular outcomes, with fewer studies investigating cognitive and psychological domains. The limited number of long-term trials restricts conclusions about the sustainability of benefits over time. Publication bias and language restrictions (English-only) may also have excluded relevant findings.

Future research should prioritize larger, long-term randomized controlled trials that explore diverse populations and settings, standardize intervention parameters, and integrate objective adherence tracking and mechanistic biomarkers to strengthen the evidence base.

## 5. Conclusions

This systematic review provides robust evidence that brief, intermittent bouts of physical activity, commonly termed exercise snacks, represent a time-efficient, feasible, and evidence-based strategy to mitigate the adverse effects of prolonged sedentary behavior. Consistent improvements were observed in metabolic regulation, vascular function, cardiorespiratory fitness, and physical function across adult and older populations. By incorporating short bouts of movement throughout the day, individuals can achieve meaningful health benefits without requiring structured exercise sessions. Exercise snacking represents a promising behavioral strategy that complements, not replaces, traditional exercise. When systematically incorporated into daily life, it may yield multi-system benefits encompassing metabolic, cardiovascular, cognitive, and psychosocial domains. Future studies should directly compare intermittent exercise-snack models with traditional continuous training to establish optimal combinations addressing metabolic, cardiovascular, and psychosocial health outcomes.

## Figures and Tables

**Figure 1 healthcare-13-03216-f001:**
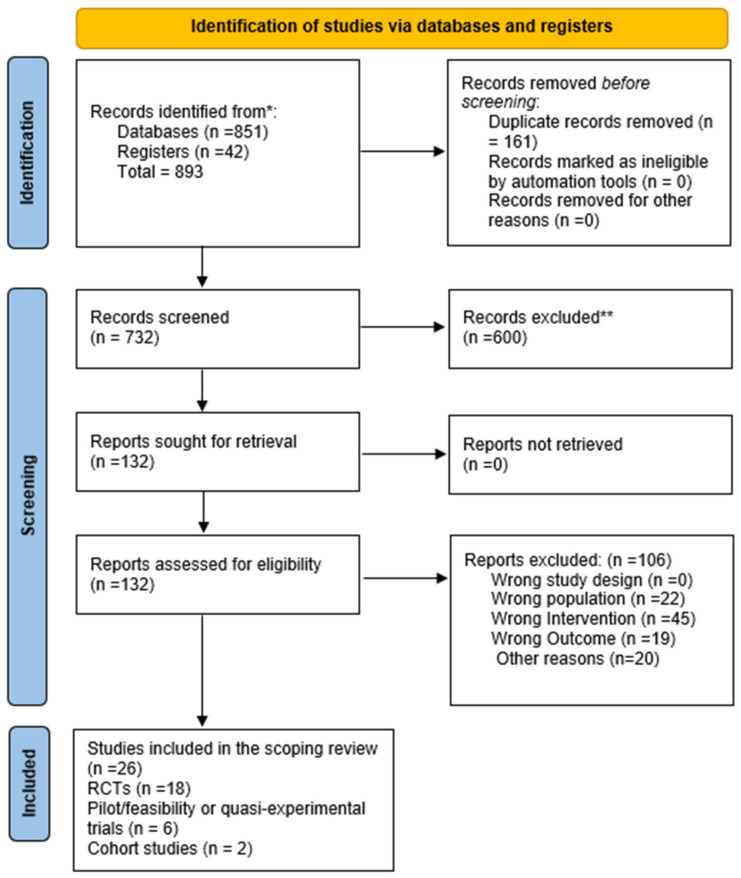
PRISMA 2020 flow diagram for new systematic reviews that included searches of databases and registers only, * the number of records identified from each database or register, ** indicate how many records were excluded.

**Table 1 healthcare-13-03216-t001:** Inclusion and Exclusion Criteria of the Review.

Criterion Type	Inclusion Criteria	Exclusion Criteria
Population	Human participants of any age (adults, older adults, clinical populations such as T2D, PCOS, obese).	Animal studies; pediatric-only studies (<18 y); studies in elite athletes only.
Intervention	Exercise snacks, activity breaks, interruptions of prolonged sitting, stair climbing snacks, home-based resistance or Tai Chi snacking.	Conventional structured exercise programs not classified as “exercise snacks”; pharmacological or dietary-only interventions.
Comparison	Control groups with uninterrupted sitting, usual care, or alternative exercise modes (e.g., MICT).	Studies without a comparator or lacking baseline/control conditions.
Outcomes	Metabolic (glucose, insulin, triglycerides), vascular (BP, FMD, CBF), fitness (VO_2_ peak, CRF), cognition, fatigue, functional outcomes (SPPB, sit-to-stand).	Outcomes unrelated to exercise/health (e.g., biomechanical modeling, unrelated psychology outcomes).
Study Design	Randomized controlled trials, randomized crossover trials, pilot RCTs, feasibility/acceptability studies, and cohort studies with relevant sedentary/exercise snack exposure.	Narrative reviews, editorials, conference abstracts, and non-peer-reviewed gray literature.
Publication Characteristics	Peer-reviewed articles published in English between 2012–2025.	Non-English language papers, theses, dissertations, and book chapters.

**Table 2 healthcare-13-03216-t002:** Risk of Bias/Quality Assessment of Included Studies.

Study	Randomization	Deviations	Missing Data	Measurement	Overall Risk
(Allison et al., 2017) [5]	Low	Low	Low	Low	Low
(Bergouignan, Latouche et al., 2016) [13]	Low	Some concerns	Low	Low	Some concerns
(Bergouignan, Legget et al., 2016) [24]	Low	Low	Low	Low	Low
(Carter et al., 2018) [21]	Low	Low	Low	Low	Low
(Zhou et al., 2025) [16]	Low	Some concerns	Low	Low	Low
(Dempsey, Larsen et al., 2016) [11]	Low	Low	Low	Low	Low
(Dempsey, Sacre et al., 2016) [12]	Low	Low	Low	Low	Low
(Diaz et al., 2017) [1]	N/A (Cohort)	Low	Low	Low	Low
(Dunstan et al., 2012) [2]	Low	Low	Low	Low	Low
(Fyfe et al., 2022) [26]	Some concerns	Low	Low	Low	Some concerns
(Jenkins et al., 2019) [6]	Low	Low	Low	Low	Low
(Larsen et al., 2014) [18]	Low	Low	Low	Low	Low
(Liang et al., 2022) [7]	Low	Some concerns	Low	Low	Low
(Liang et al., 2023) [27]	N/A (Survey)	N/A	N/A	N/A	Low
(Logan et al., 2025) [29]	Low	Low	Low	Low	Low
(Mues et al., 2025) [25]	Some concerns	Some concerns	Low	Low	Some concerns
(Peddie et al., 2013) [10]	Low	Low	Low	Low	Low
(Thosar et al., 2015) [19]	Low	Low	Low	Low	Low
(Taylor et al., 2021) [22]	Low	Low	Low	Low	Low
(Restaino et al., 2015) [20]	Low	Low	Low	Low	Low
(Western et al., 2023) [28]	Some concerns	Low	Low	Low	Some concerns
(Wennberg et al., 2016) [23]	Some concerns	Low	Low	Low	Some concerns
(Francois et al., 2014) [14]	Low	Low	Low	Low	Low
(Yin et al., 2024) [17]	Low	Low	Low	Low	Low

Note: Inter-rater agreement for inclusion and risk-of-bias assessments was κ = 0.91, indicating excellent consistency between reviewers.

**Table 3 healthcare-13-03216-t003:** Participants’ Characteristics of Included Studies.

Study (Author, Year)	Country	Population Type	Sample Size (*n*)	Age (Mean ± SD/Range)	Sex (% Male/Female)	Health Status/Condition
(Allison et al., 2017) [5]	Canada	Inactive young women	31	18–30 y	0/100	Healthy, sedentary
(Bergouignan, Latouche et al., 2016) [13]	Australia/France	Overweight/obese adults	19	35–55 y	~50/50	Overweight/obese
(Bergouignan, Legget et al., 2016) [24]	USA	Adults, sedentary workers	22	30–55 y	45/55	Sedentary, healthy
(Carter et al., 2018) [21]	UK	Healthy young adults	18	20–40 y	50/50	Healthy
(Zhou et al., 2025) [16]	China	Sedentary obese adults	60	25–45 y	40/60	Obese, otherwise healthy
(Dempsey, Larsen et al., 2016) [11]	Australia	Adults with type 2 diabetes	24	45–70 y	60/40	T2D
(Dempsey, Sacre et al., 2016) [12]	Australia	Adults with type 2 diabetes	24	45–70 y	60/40	T2D
(Brakenridge et al., 2022) [38]	Australia	Adults with T2D	Protocol only	–	–	T2D
(Diaz et al., 2017) [1]	USA (NHANES)	Community-dwelling adults	7985	≥45 y	~50/50	General population
(Dunstan et al., 2012) [2]	Australia	Overweight/obese adults	19	45–65 y	55/45	Overweight/obese
(Francois & Little, 2015) [15]	Canada	Adults with T2D	Review (no n)	–	–	T2D
(Fyfe et al., 2022) [26]	Australia	Older adults	40	65–80 y	40/60	Community-dwelling, inactive
(Jenkins et al., 2019) [6]	Canada	Young sedentary adults	24	20–30 y	50/50	Healthy sedentary
(Larsen et al., 2014) [18]	Australia	Overweight/obese adults	19	45–65 y	60/40	Overweight/obese
(Liang et al., 2022) [7]	UK/Taiwan	Older adults (COVID)	52	65–85 y	45/55	Self-isolating older adults
(Liang et al., 2023) [27]	UK/Taiwan	Older adults (survey)	200	65–85 y	45/55	Low/high-function older adults
(Logan et al., 2025) [29]	Australia	Adults with T2D	25	50–70 y	60/40	T2D
(Mues et al., 2025) [25]	Germany	Middle-aged office workers	48	40–55 y	50/50	Sedentary, cognitively healthy
(Peddie et al., 2013) [10]	New Zealand	Healthy normal-weight adults	70	20–35 y	~50/50	Healthy
(Thosar et al., 2015) [19]	USA	Young men	12	18–30 y	100/0	Healthy
(Taylor et al., 2021) [22]	Australia	Women with PCOS	28	25–40 y	0/100	PCOS
(Restaino et al., 2015) [20]	USA	Healthy adults	15	18–30 y	55/45	Healthy
(Western et al., 2023) [28]	UK	Pre-frail older adults	34	70–85 y	35/65	Pre-frail, memory clinic
(Wennberg et al., 2016) [23]	Sweden	Overweight adults	25	40–60 y	50/50	Overweight, sedentary
(Francois et al., 2014) [14]	New Zealand	Adults with insulin resistance	12	40–65 y	60/40	Insulin resistant
(Yin et al., 2024) [17]	China	Inactive adults	50	20–40 y	50/50	Healthy sedentary

**Table 4 healthcare-13-03216-t004:** Summary of Health Domains Influenced by Exercise Snacks (2012–2025).

Health Domain	Representative Outcomes Observed	Direction of Effect	Typical Evidence Source
Metabolic Health	↓ post-prandial glucose, ↓ insulin, ↓ triglycerides; improved glucose tolerance	Positive	Randomized crossover and RCTs [2,11,14]
Cardiovascular Function	↓ resting and post-prandial blood pressure; ↑ endothelial FMD and cerebral blood flow	Positive	Laboratory and clinical trials [19,21,22]
Functional/Musculoskeletal Fitness	↑ lower-limb strength, ↑ sit-to-stand and SPPB scores	Positive	Pilot and home-based RCTs [26,28]
Cognitive/Psychological Performance	↑ alertness, ↓ fatigue, ↑ mood, improved acute cognition	Positive	Experimental and workplace studies [24,25]
Feasibility/Behavioral Adherence	High acceptability, ≥ 80% adherence; sustainable daily integration	Positive	Feasibility and survey data [7,27]

**Table 5 healthcare-13-03216-t005:** Characteristics of Studies Investigating the Effects of Exercise Snacks on Metabolic, Cardiovascular, Cognitive, and Functional Health Outcomes (2012–2025).

Author & Year	Aim	Population	Intervention	Comparison	Outcome	Study Design	Test Results
(Allison et al., 2017) [5]	Examine whether brief, intense stair climbing improves cardiorespiratory fitness	Inactive young women	3 × 20-s stair climbing bouts/day for 6 weeks	Control (no training)	Cardiorespiratory fitness (VO_2_ peak)	Randomized trial	VO_2_ peak vs. control
(Bergouignan, Latouche et al., 2016) [13]	Assess molecular pathways from frequent sedentary interruptions	Overweight/obese adults	Frequent walking breaks	Prolonged sitting	Glucose uptake pathways	Randomized crossover	Improved insulin-stimulated glucose uptake
(Bergouignan, Legget et al., 2016) [24]	Evaluate psychological and behavioral responses to sitting interruptions	Adults	5-min walking every hour	Uninterrupted sitting	Energy, mood, cravings, cognition	Randomized crossover	Energy, cravings, improved mood
(Carter et al., 2018) [21]	Investigate impact of walking breaks on cerebral blood flow	Healthy adults	5-min light walking every 30 min	Prolonged sitting	Cerebral blood flow (CBF)	Randomized crossover	Walking breaks prevented decline in CBF
(Zhou et al., 2025) [16]	Effect of exercise snacks on body composition and metabolomics	Sedentary obese adults	Exercise snacks intervention	Uninterrupted sitting	Body composition, metabolomics	Randomized controlled trial	Improved composition and plasma metabolomics
(Dempsey, Larsen et al., 2016) [11]	Interrupting sitting with walking/resistance in T2D	Adults with T2D	3-min walking or resistance breaks every 30 min	Uninterrupted sitting	Glucose, insulin, triglycerides	Randomized crossover	Postprandial glucose, insulin, TGs
(Dempsey, Sacre et al., 2016) [12]	Impact of activity breaks on BP and noradrenaline	Adults with T2D	3-min light walking or resistance every 30 min	Uninterrupted sitting	Blood pressure, noradrenaline	Randomized crossover	BP, Noradrenaline
(Brakenridge et al., 2022) [38]	Protocol for OPTIMISE trial	Adults with T2D	Sitting less, moving more program	Usual care	Metabolic and brain health	RCT protocol	Planned outcomes, not reported
(Diaz et al., 2017) [1]	Association between sedentary patterns and mortality	US adults (NHANES)	Model replacing sedentary with activity	Prolonged sedentary	All-cause mortality	Cohort study	More breaks mortality risk
(Dunstan et al., 2012) [2]	Effect of breaking sitting on glucose/insulin	Overweight adults	2-min light/mod walking every 20 min	Uninterrupted sitting	Postprandial glucose, insulin	Randomized crossover	Glucose & insulin AUC
(Francois & Little, 2015) [15]	Evaluate HIIT safety and effectiveness in T2D	Adults with T2D	High-intensity interval training (exercise snacks)	Usual activity	Glycemic control, safety	Review/clinical evidence	HIIT safe and effective
(Fyfe et al., 2022) [26]	Feasibility of resistance exercise snacking in older adults	Community-dwelling older adults	Home-based resistance snacks (pragmatic RCT)	Control	Physical function, feasibility	Pilot RCT	Feasible and acceptable
(Jenkins et al., 2019) [6]	Stair climbing exercise snacks and fitness	Young adults	3 Ã/day vigorous stair climbing for 6 weeks	Control	Cardiorespiratory fitness	Randomized trial	VO_2_ peak
(Larsen et al., 2014) [18]	Breaking up sitting and blood pressure	Overweight/obese adults	Walking breaks	Uninterrupted sitting	Resting blood pressure	Randomized crossover	Resting BP
(Liang et al., 2022) [7]	Feasibility of home-based exercise/Tai Chi snacks	Older adults (COVID isolation)	Remotely delivered exercise & Tai Chi snacks	None	Feasibility, acceptability	Pilot trial	Well accepted
(Liang et al., 2023) [27]	Acceptability of exercise/Tai Chi snacks	UK & Taiwanese older adults	Home-based exercise and Tai Chi snacks	None	Acceptability	Cross-cultural survey	High acceptability in both groups
(Logan et al., 2025) [29]	Interrupting sitting effects on incretin hormones	Adults with T2D	Light walking breaks	Prolonged sitting	GIP, GLP-1 responses	Randomized crossover	GIP, GLP-1 unchanged
(Mues et al., 2025) [25]	Workplace exercise snacks and cognition	Sedentary middle-aged adults	Short exercise snacks during workday	Usual work routine	Cognitive performance	Randomized pilot trial	Acute cognition, feasible
(Peddie et al., 2013) [10]	Compare sitting breaks vs. single exercise bout	Healthy adults	1–2 min walking every 30 min	Single 30-min bout; uninterrupted sitting	Postprandial glucose, insulin	Randomized crossover	Breaks better at glucose, insulin
(Thosar et al., 2015) [19]	Effect of sitting and breaks on endothelial function	Young adults	Light walking breaks during sitting	Prolonged sitting	Endothelial function (FMD)	Randomized crossover	Breaks prevented decline in FMD
(Taylor et al., 2021) [22]	Effect of sitting breaks in PCOS women	Women with PCOS	Interrupting sitting with activity	Prolonged sitting	Endothelial function	Randomized crossover	Improved endothelial function
(Restaino et al., 2015) [20]	Vascular effects of prolonged sitting	Healthy adults	Leg movement vs. no movement	Prolonged sitting	Micro/macrovascular dilator function	Experimental crossover	Vascular function with sitting
(Western et al., 2023) [28]	28-day exercise snacking in pre-frail older adults	Pre-frail memory clinic patients	Daily home-based resistance snacks	Usual routine	Physical function (SPPB, sit-to-stand)	Pilot pre-post	Improved lower-limb function
(Wennberg et al., 2016) [23]	Breaking sitting and fatigue/cognition	Overweight adults	Light walking breaks	Prolonged sitting	Fatigue, cognition	Pilot crossover	Fatigue, mixed cognition effects
(Francois et al., 2014) [14]	Pre-meal exercise snacks and glycemic control	Adults with insulin resistance	Short HIIT snacks before meals	Continuous exercise; sitting	Postprandial glucose, insulin	Randomized crossover	Snacks more effective than continuous exercise
(Yin et al., 2024) [17]	Compare exercise snacks vs. MICT on CRF/fat oxidation	Inactive adults	Exercise snacks for 6 weeks	MICT training	CRF, fat oxidation	Randomized controlled trial	Snacks improved CRF, not maximal fat oxidation

T2D = Type 2 Diabetes; FMD = Flow-Mediated Dilation; CRF = Cardiorespiratory Fitness; MICT = Moderate-Intensity Continuous Training; CBF = Cerebral Blood Flow; SPPB = Short Physical Performance Battery; TGs = Triglycerides; GIP = Glucose-Dependent Insulinotropic Polypeptide; GLP-1 = Glucagon-Like Peptide-1.

**Table 6 healthcare-13-03216-t006:** GRADE Evidence Summary.

Outcome	No. of Studies	Design	Risk of Bias	Inconsistency	Indirectness	Imprecision	Publication Bias	Certainty
Metabolic control (glucose/insulin)	12	RCTs & crossovers	Low	Low	Low	Low	Low	Moderate
Cardiorespiratory fitness	5	RCTs	Low	Low	Low	Low	Low	High
Vascular health (BP, FMD, CBF)	7	RCTs	Low	Some concerns	Low	Low	Low	Moderate
Cognitive outcomes	4	Pilot RCTs	Some concerns	High	Moderate	High	Some concerns	Low
Older adult functional outcomes	5	RCTs & pilots	Low	Low	Low	Low	Low	High

**Table 7 healthcare-13-03216-t007:** Measurement Protocols for Outcomes in Included Studies.

Study (Author, Year)	Outcome(s) Measured	Measurement Protocol/Instrument Used
(Allison et al., 2017) [5]	Cardiorespiratory fitness (VO_2_ peak)	Graded treadmill exercise test with indirect calorimetry
(Bergouignan, Latouche et al., 2016) [13]	Glucose uptake pathways	Muscle biopsies; insulin- and contraction-stimulated glucose uptake assays; molecular pathway analysis
(Bergouignan, Legget et al., 2016) [24]	Energy, mood, cravings, cognition	Self-reported visual analog scales; validated questionnaires
(Carter et al., 2018) [21]	Cerebral blood flow (CBF)	Transcranial Doppler ultrasound (middle cerebral artery velocity)
(Zhou et al., 2025) [16]	Body composition, metabolomics	DXA for composition; plasma metabolomic profiling via LC-MS
(Dempsey, Larsen et al., 2016) [11]	Postprandial glucose, insulin, TGs	Capillary/venous blood sampling every 30–60 min for 7 h; enzymatic assays
(Dempsey, Sacre et al., 2016) [12]	Blood pressure, noradrenaline	Automated oscillometric BP; plasma noradrenaline via HPLC
(Brakenridge et al., 2022) [38]	Planned metabolic & brain outcomes	Protocol—planned HbA1c, fasting glucose, MRI brain scans, cognitive battery
(Diaz et al., 2017) [1]	Mortality, sedentary patterns	Accelerometer-based sedentary assessment; mortality from NHANES linkage
(Dunstan et al., 2012) [2]	Postprandial glucose, insulin	Venous blood samples during 5-h meal test; AUC calculations
(Francois & Little, 2015) [15]	Glycemic control, safety	Narrative/clinical evidence (varied methods across HIIT trials)
(Fyfe et al., 2022) [26]	Physical function, feasibility	30-s chair stand, timed up-and-go, 6-min walk test; feasibility via adherence logs & surveys
(Jenkins et al., 2019) [6]	Cardiorespiratory fitness	VO_2_ peak test via incremental cycle ergometer
(Larsen et al., 2014) [18]	Resting blood pressure	Automated BP monitor (average of repeated seated measures)
(Liang et al., 2022) [7]	Physical function, acceptability	30-s sit-to-stand, balance tests; surveys on feasibility/acceptability
(Liang et al., 2023) [27]	Acceptability	Semi-structured surveys/interviews
(Logan et al., 2025) [29]	Incretin hormones (GIP, GLP-1)	Venous blood sampling post-meal with ELISA-based assays
(Mues et al., 2025) [25]	Cognitive performance	Computerized cognitive tests (working memory, reaction time, Stroop task)
(Peddie et al., 2013) [10]	Postprandial glucose, insulin	Capillary blood glucose; insulin ELISA during standardized meal test
(Thosar et al., 2015) [19]	Endothelial function (FMD)	Brachial artery FMD by high-resolution ultrasound
(Taylor et al., 2021) [22]	Endothelial function in PCOS	FMD of brachial artery; reproductive hormone profiling
(Restaino et al., 2015) [20]	Micro/macrovascular dilation	Ultrasound-based FMD; microvascular function via local heating/shear stimulus
(Western et al., 2023) [28]	Physical function	Short Physical Performance Battery (SPPB); 5-times sit-to-stand
(Wennberg et al., 2016) [23]	Fatigue, cognition	Self-reported fatigue scales; computerized attention/working memory tests
(Francois et al., 2014) [14]	Glycemic control (pre-meal snacks)	OGTT-like protocol; repeated postprandial blood draws (glucose, insulin)
(Yin et al., 2024) [17]	CRF, fat oxidation	Incremental treadmill VO_2_ peak test; indirect calorimetry for fat oxidation rates

## Data Availability

No new data were generated or analyzed in this study. All data supporting the findings of this systematic review are available within the article. The datasets extracted and analyzed were obtained from previously published studies cited in the review.

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
