# Peer review of "Exercise Snacks as a Strategy to Interrupt Sedentary Behavior: A Systematic Review of Health Outcomes and Feasibility"

_healthcare, 2025, doi:10.3390/healthcare13243216_

Round 1
Reviewer 1 Report
Comments and Suggestions for Authors
Please see attached

Author Response
Dear Reviewer 1,
Time is extremely precious for every person. The authors sincerely thank you for taking the time out of your life to study and analyze our manuscript and for offering us suggestions and recommendations.
Attached is the point-by-point response to your comments.
Thank you

Reviewer 2 Report
Comments and Suggestions for Authors
Dear Authors
As one of the reviewers, I express my personal scientific opinion on your work. I would like to reassure you that I was trying to be positive and constructive but particularly as fair and honest as possible to your work. First of all, well done for the conceptualization of the topic and the for the time spent to accomplish this massive task. The good and logical flow in the Introduction and the presentation of the results are appreciated. I should also note that the originality of the study and the work done on tables and figure are all positive points. However, some more explanations, clarifications mainly, are required in Method’s section. In addition, some aspects related to the structure of the report need further consideration.
Please accept my judgment with a positive and constructive way.
Abstract:
- Lines 38-40: “A total of 26… type 2 diabetes, or PCOS”; In my point of view, the whole sentence should be removed above, to the “Methods” part of the abstract.
Introduction:
- Very well organized and written Introduction section.
Methods:
- Please improve the quality of Figure 1.
- Although in your Table 1, you have nicely reported the Inclusion and Exclusion Criteria of the review, what were specifically the cut-offs criteria for defining a study as “wrong” population (n=22) and intervention (n=45) and which criteria also did you use for reaching the conclusion that a study had “wrong” outcome n=19)? Please clarify in some more details. If these studies had wrong population, intervention and outcome, how did they overcome the review process and be published.
- What about the other reasons for excluding a study? Roughly, which are the other reasons?
- Overall (comments, by reporting “wrong” studies, did you mean perhaps “inappropriate” to the current review studies?
Results:
- Lines 268-277: In my point of view, the whole paragraph should be removed to the methods section.
Discussion:
- How much time for “exercise snacks”? And what about intensity of exercise for these snacks?
- Please offer to the readers a specific direction (practical application) of how practically will be organized in order to implement this “snacks” into their daily life, either at work or at home and social activities. Any motivational tips?
- A much of text overlapping/repetition exists between the Introduction and Discussion sections. Please find the way to reduce it as much as possible for making your report more attractive and interesting.
Conclusion:
- Lines 424-456: I found the conclusions too long and exhausting. Please cut it off down in one small, hit to the point paragraph. You could use a similar approach as the abstract’s conclusions.
- Furthermore, in my point of view, the last paragraph of the report/conclusion should be cut off and if it is possible please create, above the conclusion section, a new subsection which could be called it “Study limitations”.
- Lines 424-426: “This systematic review synthesised 26… sedentary behavior”, Please eliminate the whole sentence for avoiding repetition.
Author Response
Dear Reviewer 2,
Time is extremely precious for every person. The authors sincerely thank you for taking the time out of your life to study and analyze our manuscript and for offering us suggestions and recommendations.
Attached is the point-by-point response to your comments.
Thank you

Reviewer 3 Report
Comments and Suggestions for Authors
This article is a comprehensive and timely systematic review that focuses on evaluating the efficacy and feasibility of "Exercise Snacks" against prolonged sedentary behaviors. This topic is of great importance in the fields of exercise physiology and public health.
The main strength of the paper is its broad coverage of outcomes (metabolic, cardiovascular, cognitive, and functional) and the inclusion of feasibility studies.
However, the manuscript has major deficiencies in several key aspects of systematic review methodology and scientific reporting that must be addressed prior to final acceptance.

Author Response
Dear Reviewer 3,
Time is extremely precious for every person. The authors sincerely thank you for taking the time out of your life to study and analyze our manuscript and for offering us suggestions and recommendations.
Attached is the point-by-point response to your comments.
Thank you

Reviewer 4 Report
Comments and Suggestions for Authors
Dear authors, thank you for the opportunity to read and learn from your work. I believe that this work provides valuable information on "exercise snacks". This proposal highlights multiple health benefits in all its dimensions. Below, you will find a series of reviews that may improve your work. Initially, you will find them in a more general form and later in a more specific form associated with each section.
- The number of works from the last five years is limited. It would be advisable to update the information in the manuscript with more current references that address the variables of the work in whole or in part.
- More information on health in all its dimensions.
- Detail the procedure more specifically in terms of materials and methods.
- The results section could be reorganised and improved.
- Eliminate repetitions in the discussion and expand on a more in-depth analysis of the possibilities of traditional exercise vs. exercise snacks.

Author Response
Dear Reviewer 4,
Time is extremely precious for every person. The authors sincerely thank you for taking the time out of your life to study and analyze our manuscript and for offering us suggestions and recommendations.
Attached is the point-by-point response to your comments.
Thank you

Round 2
Reviewer 1 Report
Comments and Suggestions for Authors
I am happy with the major revisions. I believe the manuscript is now ready for publication.
Regards
Reviewer 4 Report
Comments and Suggestions for Authors
Thank you very much for your response. The manuscript has improved significantly. All requests have been addressed appropriately.